# Role of Melatonin in Cancer: Effect on Clock Genes

**DOI:** 10.3390/ijms24031919

**Published:** 2023-01-18

**Authors:** César Rodríguez-Santana, Javier Florido, Laura Martínez-Ruiz, Alba López-Rodríguez, Darío Acuña-Castroviejo, Germaine Escames

**Affiliations:** 1Biomedical Research Center, Health Sciences Technology Park, University of Granada, 18016 Granada, Spain; 2Department of Physiology, Faculty of Medicine, University of Granada, 18071 Granada, Spain; 3CIBER de Fragilidad y Envejecimiento Saludable (CIBERFES), Instituto de Salud Carlos III, Instituto de Investigación Biosanitaria Ibs., 18016 Granada, Spain

**Keywords:** melatonin, cancer, clock genes, circadian rhythms, SIRT1, *c-Myc*

## Abstract

The circadian clock is a regulatory system, with a periodicity of approximately 24 h, that generates rhythmic changes in many physiological processes. Increasing evidence links chronodisruption with aberrant functionality in clock gene expression, resulting in multiple diseases, including cancer. In this context, tumor cells have an altered circadian machinery compared to normal cells, which deregulates the cell cycle, repair mechanisms, energy metabolism and other processes. Melatonin is the main hormone produced by the pineal gland, whose production and secretion oscillates in accordance with the light:dark cycle. In addition, melatonin regulates the expression of clock genes, including those in cancer cells, which could play a key role in the numerous oncostatic effects of this hormone. This review aims to describe and clarify the role of clock genes in cancer, as well as the possible mechanisms of the action of melatonin through which it regulates the expression of the tumor’s circadian machinery, in order to propose future anti-neoplastic clinical treatments.

## 1. Introduction

Disruptions in circadian rhythms have been linked to mammalian tumorigenesis in numerous studies [1]. At the molecular level, circadian rhythms are generated by a system of interlocking autoregulatory transcriptional/translational feedback loops, while melatonin (N-acetyl-5-methoxytryptamine, aMT) is the principal hormone that regulates circadian clock genes.

Although melatonin is produced by the pineal gland, higher concentrations are also produced in other tissues [2]. The synthesis of pineal melatonin is associated with light/dark photoperiods, with increasing levels produced at night [3]. This facilitates the chronobiotic regulation of numerous activities, such as immunomodulatory, anti-inflammatory, antioxidant and vasoregulatory activities [4,5,6,7].

Numerous studies have found a strong correlation between reduced melatonin levels and increased risk of tumor development [8,9,10,11,12]. The oncostatic properties of melatonin have been extensively studied. Although this hormone’s mechanisms of action in cancer remain unclear, its connection with the circadian clock may be key to a better understanding of these mechanisms.

This review therefore emphasizes the connection between the disruption of circadian rhythms, cancer and the role of melatonin in regulating clock genes.

## 2. General Mechanism of the Circadian Clock

In mammals, the circadian clock acts as a regulatory system, with a periodicity of approximately 24 h. This process is modulated by endogenous factors such as genetic or endocrine secretions, behavioral factors such as feeding/fasting, as well as external oscillating signals such as light/darkness and temperature cycles. All these inputs generate rhythmic changes in many physiological processes, including the endocrine system, the cell cycle, DNA damage repair, metabolism and the sleep/wake cycle [13].

The circadian clock is controlled by a central circadian pacemaker located in the suprachiasmatic nucleus (SCN) of the hypothalamus, which, in turn, controls the peripheral oscillators, located in practically all cells in the body [14]. The synchronization of the cells in different tissues requires the precise coordination of the circadian clock. However, although circadian rhythms are fundamental to the correct physiological functioning of the organism, these mechanisms can be lost or weakened due to different factors, such as aging or various pathologies [15].

At the molecular level, the central circadian clock is composed of a small number of genes whose expression forms a time-delayed transcription–translation feedback loop. The activating arm of the loop is composed of circadian locomotor output cycle kaput (CLOCK)/neuronal PAS domain protein 2 (NPAS2) and brain and muscle aryl hydrocarbon receptor nuclear translocator-like protein 1 (BMAL1), which form a heterodimer and connect with its promoter at CACGTG E-box sequences. The binding to the E-box by this heterodimer induces clock-controlled gene (CCG) expression. This facilitates the transcription of numerous genes, including the Cryptochrome (*cry1* and *cry2*) and Period (*per1*, *per2* and *per3*) genes. The CRY and PER proteins form a repressor complex that is translocated into the nucleus during the evening and physically interacts with the CLOCK/NPAS2:BMAL1 heterodimer in order to repress their own transcription. Subsequently, these proteins, which increase and accumulate in the cytoplasm, are phosphorylated by cysteine kinase 1 ε/δ (CK1 ε/δ) during the night for its degradation. They are targeted for ubiquitination by specific E3 ligases and are eventually degraded by the proteasome, thus further increasing their synthesis at the beginning of the day. The waxing and waning of this transcriptional feedback loop, which takes ~24 h to complete, represents the core mechanism of the circadian clock in mammals [16,17] (Figure 1).

In turn, *Bmal1* is regulated by a second loop composed of the activator orphan retinoic acid receptor-related alpha (RORα), and the repressor, nuclear reactors reverse strand of protein ERB alpha (REV-ERBα) (Figure 1). At the same time, clock genes can be further modulated by epigenetic modifications and miRNA, as well as by post-transcriptional and post-translational modifications [17,18,19].

## 3. Clock Genes and Cancer

Alterations in clock genes are associated with different types of human cancer [20,21,22,23]. However, it is not understood how circadian disruption is associated with serious adverse health outcomes, including carcinogenesis. 

The World Health Organization currently recognizes night shift work, which involves circadian disruption, as a probable carcinogen, classified as type 2A by the Agency for Research on Cancer (IARC) [24]. Since the 1980s, numerous epidemiological studies have linked an increased risk of different cancers, such as breast and prostate cancer, with night shift work by employees such as nurses and flight attendants on transatlantic flights [1,10,12,25]. It has been suggested that sleep deprivation, the light-induced suppression of melatonin and lifestyle changes are important mechanisms that could explain the possible link between shift work and cancer risk [24]. Moreover, studies of genetic variations in the circadian pathway in human clock genes have reported that the disruption of circadian rhythms is related to an increased risk of cancer [26]. In support of these findings, a significant relationship has been established between different types of cancer, such as breast, lung and prostate cancer, and variations in *Bmal1, Clock, RORα* and *RORβ*. This is due to the fact that clock genes regulate between approximately 50% and more than 80% of mammalian genome genes, including the tumor suppressor gene *p53* and the oncogene *c-Myc* [27,28]. The disruption in the circadian clock is therefore involved in tumor development by altering the expression of genes involved in fundamental functions, such as the cell cycle, apoptosis, metabolism and energy, DNA repair, tumor immunity and metastasis [29]. In addition, the polymorphisms of other clock genes specific to each tumor type have been identified, suggesting that certain circadian genes might be more important than others in terms of predisposition to different types of cancer [26].

Intriguingly, given that cancer prognosis and survival have been associated with the level of circadian disruption in patient tumor tissues [30], clock genes, such as *Bmal1*, *Per1* and *npas2,* could be considered potential prognostic biomarkers in certain cancers [31,32].

Nevertheless, other studies found no evidence of a relationship between interruptions in the circadian rhythm in night shift workers and carcinogenesis [33]. Recently, in a meta-analysis of 57 observational studies, no evidence of a relationship was reported between night shift work and an increased risk of cancer, especially in breast, prostate, ovarian, pancreatic, colorectal, non-Hodgkin’s lymph and stomach cancers [34]. However, these differences in results can be explained by the methodological differences of the different epidemiological studies, as well as in the statistical tools used [35]. Therefore, common criteria to clarify the relationship between an alteration in circadian rhythms and carcinogenesis are clearly required.

### 3.1. Clock Genes, Proliferation and Apoptosis

Malignant tumors are characterized by uncontrolled cell proliferation, partly due to a loss of control of cell cycle events caused by clock gene dysregulation [36]. Clock genes affect many biological pathways, including those involved in cell proliferation and apoptosis, by controlling the expression of cell cycle genes [29]. In addition, substantial evidence shows that progression through the cell cycle occurs at specific times of the day/night cycle, suggesting that one function of the circadian clock system is to control this fundamental process [37]. It is remarkable that circadian rhythms share some common features with the cell cycle, while disruptions in circadian clocks have been found to be related to carcinogenesis. For example, the BMAL1-CLOCK/NPAS2 heterodimer has been reported to repress *c-Myc*, an oncogene that contributes to the genesis of many human cancers, and whose protein expression is closely correlated with cell proliferation rates [30,38]. REV-ERBα and RORα have also been reported to regulate the expression of *p21*, a CCG that negatively regulates cell cycle progression [37]. Overexpressed PER1 also induces *c-Myc* and suppresses *p21,* while other studies have reported that PER1 inhibits *Cyclin B1*, *Cdc2* and *Wee1* expression, leading to a decrease in cancer cell proliferation [39]. This discrepancy may be due to differences in the methods used or in the characteristics of the cell lines used in the experiments [40]. Nevertheless, these data indicate that many genes crucial to the cell cycle are under the control of clock genes that are aberrantly expressed in many tumor tissues.

It has also been reported that many crucial genes involved in cell proliferation and apoptosis have periodic patterns of expression. These genes, which oscillate during a 24-h cycle, include the proliferation gene *Ki-67*, the tumor suppressor gene *p53*, the proto-oncogene *Mdm2* and the apoptotic-related proteins BAX and BCL-2 [29]. PER1 and PER2 have been reported to be mainly associated with the upregulation of *Ki-67*, *Mdm2* and *Bax,* as well as with the downregulation of *Bcl*-*2*, c-Myc and *p53,* in lung, pancreatic, hepatocellular and oral carcinoma cell lines [41,42,43]. In addition, there is a strong connection between *Bmal1* and *Per2* regulation and the PI3K/mTOR signaling pathway, one of the most frequently activated signaling pathways in tumorigenesis and the progression of cancer. For example, BMAL1 depletion leads to cell cycle disruption, which results in a substantial increase in the apoptotic cell population, and to the acceleration of cell invasion [44]. The PI3K/mTOR pathway has a major effect on the regulation of processes such as autophagy, proliferation and apoptosis, which sequentially affects the occurrence and development of cancer [44,45,46].

Many signaling pathways cooperate with clock genes involved in tumorigenesis; the altered expression of these genes can modify a range of downstream CCGs and tumor-related genes, which impacts tumor cell proliferation, apoptosis, migration and invasion.

Collectively, these data suggest that the circadian clock can control cell proliferation and apoptosis at multiple levels and that disruption of the circadian system is linked to tumor cell growth.

### 3.2. Clock Genes and Metastasis

The relationship between metastatic disease progression and circadian rhythms has been poorly characterized [47,48,49]. However, a large number of studies have demonstrated the participation of clock genes in metastasis. For example, low PER1/PER2 expression in different types of cancer, such as breast [50], glioma [51], gastric [52] and non-small cell lung cancer [53], is closely related to the development and metastasis of tumors. Moreover, much evidence has identified BMAL1 as a key element in metastasis in breast cancer and glioblastoma. BMAL1 regulates the expression and activity of matrix metalloproteinase 9 (MMP9), which controls cell migration and invasion [47,48]. MMP9, which is involved in the degradation of the tumor extracellular matrix, is a mediating factor with regard to the local invasion and distant metastasis of tumor cells. In human colorectal cancer, BMAL1 has been shown to induce metastasis by stimulating exosome secretion [54]. Exosomes derived from primary tumors have been shown to alter the microenvironment of secondary organs in order to facilitate the colonization and growth of metastatic tumors, whose quantity is dependent on circadian rhythms [54].

### 3.3. Clock Genes and Tumor Immunity

Circadian rhythms are also involved in the mammalian immune system, which involves various populations of immune cells, such as monocytes, natural killer cells, dendritic cells (DCs) and T and B lymphocytes, as well as responses to signals and their defensive functions, including cytokine levels [55,56]. Thus, when circadian homeostasis is disrupted, deregulation of the immune system produces immune suppression and the accelerated development of tumors. BMAL1 is the principal mediator of the circadian control of the immune system and also promotes anti-inflammatory states. Downregulation of BMAL1 has been found in hematologic malignancies such as diffuse large B-cell lymphoma, as well as acute lymphocytic and myeloid leukemias [57]. Deletion of BMAL1 affects the development of B lymphocytes [58]. In addition, a recent investigation by Wang, C. et al. [56] showed that, in murine models with the specific inhibition of *Bmal1* in T and dendritic cells, regardless of the time at which melanoma cells were inoculated, the tumor volume was similar after 14 days of the experiment. However, in wild-type mice, significant differences in tumor growth were observed depending on the time of engraftment. Therefore, BMAL1 and cell-autonomous circadian oscillations in both DCs and T cells are critical for time-of-day differences in tumor volume.

BMAL1, CLOCK, REV-ERBα and RORα have also been reported to regulate immune functions and inflammation by modulating CCGs that encode a variety of proteins, including cytokines, chemokines and receptors [29]. However, the central pacemaker is also modulated by immune factors such as proinflammatory cytokines and IL-1/6, as well as by anti-inflammatory drugs, at the molecular and cellular levels, which results in the subsequent alteration of clock genes. In conclusion, the circadian clock and immune system can be said to exert bidirectional control [59].

### 3.4. Clock Genes and Chronotherapy

The toxicity, efficacy and even the pharmacokinetics or metabolism of a drug can vary with the time of day, depending on its mechanism of action [60]. In recent years, special attention has been paid to the administration of anticancer drugs depending on circadian rhythmicity in order to maximize efficacy and to reduce side effects. For example, there is evidence that, in synchronized esophageal cancer cells, DNA damage induced by cisplatin is greater when coinciding with lower levels of PER2 [61]. In in vivo melanoma models, mice treated at night showed a higher rate of cisplatin–DNA adduct removal and less toxicity than those treated in the morning, which coincides with maximal global and gene-specific repairs. Interestingly, differences in the effects of this treatment were not observed in *Per1/2* knockout mice [62].

Chronotherapy has improved the implantation of medication for other pathologies, such as asthma [60,63], osteoarthritis and rheumatoid arthritis [64]. However, to date, studies carried out to evaluate chronochemotherapy for some types of cancer are contradictory. For example, chronotherapy in the case of ovarian cancer has shown no beneficial effect [65], while male patients with colorectal cancer lived significantly longer [66].

The main problem concerning the clinical application of chronotherapy is the need for studies of the circadian gene expression profiles of specific cancers. In addition, non-invasive methods to assess the circadian parameters of cancers need to be developed. Thus, although the use of chronotherapy in cancer treatment is an interesting avenue of study, more in-depth research is necessary [67].

Therefore, given all these data, the circadian clock clearly plays a fundamental role in tumor pathogenesis. Alterations in clock gene expression correlate with alterations in DNA replication, DNA repair and responses to DNA damage in the metabolism and in senescence. The control of cell proliferation and apoptosis is also lost, metastasis spreads, the immune system is altered and even drug resistance increases. Thus, the “repair” of circadian rhythms has been shown to be a possible therapeutic target against this disease. However, due to the great complexity of the pathways involved, further research to clarify the role of circadian clocks in cancer is required.

## 4. Melatonin and Circadian Rhythms

Melatonin, which is the principal hormone produced in the pineal gland following a circadian rhythm, is also produced in other tissues [2]. However, unlike pineal melatonin, extrapineal melatonin does not follow a circadian rhythm, and, to date, it is not known whether its synthesis is controlled by clock genes [68,69]. Pineal melatonin production is controlled by the suprachiasmatic nucleus (SCN), which transforms the photoperiodic signal into an endocrine signal (Figure 2) [69]. During the night, the polysynaptic pathway that establishes the SCN in the pineal gland gives rise to the stimulation of pinealocytes, which culminates in the expression of arylalkylamine N-acetyltransferase (AANAT), one of the key enzymes in the production of melatonin [70]. Once melatonin is synthesized in the pineal gland, it is rapidly released to the cerebrospinal fluid and blood due to its high diffusibility. As a result, the melatonin levels in both the fluid and blood have a circadian rhythm, which typically peaks at night [71]. All these rhythmic variations, including endocrine and non-endocrine rhythms, during approximately 24 h have a chronobiotic impact on the organism [3].

Melatonin, in turn, affects the SCN and, therefore, regulates the circadian phase to maintain rhythmic stability, although the way in which this is done is unclear. Melatonin membrane receptors have been identified in the SCN of vertebrates, while signal transduction pathways are involved in both MT1 and MT2 in order to induce an increase in the expression of the two clock genes, *Per1* and *Per2* [71,72]. Melatonin also has an acute inhibitory effect on neuronal firing through the inhibition of glutamatergic activity, independently of MT1 and MT2 [73]. In addition, melatonin also binds other proteins, such as calmodulin, calreticuline and quinone reductase II (previously called MT3), leading to various regulatory effects [7,74].

This hormone is also able to change the phase in the expression of *Bmal1* and *Rev-erbα* [75,76]. It has been hypothesized that melatonin maintains the regulation of the circadian machinery through the ubiquitin–proteasome system (explained below). Thus, the effect of melatonin on the central pacemaker could act as a regulator of the circadian clock by regulating the correct time and rhythmic amplitude [69,77].

Therefore, pineal melatonin is a chronobiotic molecule that plays a major role in the coordination of circadian rhythmicity. Functionally, pineal melatonin produces a host of effects that can be controlled by the SCN and may also have a direct impact on numerous peripheral organs. Thus, melatonin plays multiple roles in the complex circadian system by influencing the phases of the central pacemaker, as well as of peripheral oscillators. In particular, pineal melatonin is involved in sleep promotion [78]. At the molecular level, increased melatonin levels are correlated with BMAL1 and CLOCK expression at night, which generates circadian rhythms through the transcriptional/translational feedback loop described above [79]. Additionally, melatonin has been shown to modulate, either positively or negatively, the expression of most central oscillator genes. Nevertheless, the mechanisms by which melatonin participates in the expression of circadian clock components are unclear.

Alterations in pineal melatonin levels, caused mainly by light exposure at night, can disrupt the circadian system and have, consequently, been related to numerous pathologies, including carcinogenesis [12].

## 5. Melatonin, Clock Genes and Cancer

Numerous studies have demonstrated the relationship between disrupted clock genes, cancer and decreased levels of melatonin. Moreover, the suppression of melatonin production is associated with an increased incidence of cancer [12,80], which could be correlated with a loss in the regulation of the circadian machinery. Additionally, numerous studies have described a large number of oncostatic effects of this hormone [74,81,82], some of which could be mediated through its effects on circadian rhythms.

Xiang, S. et al. [83] have described how the use of melatonin in MCF7 breast cancer cells (1 nM) in vitro represses RORα transactivation through the MT1 receptor activation pathway, leading to the suppression of *Bmal1* mRNA expression. In addition, an increase in *Per2* expression was observed following treatment with melatonin after 20 h. Per2 is an important tumor suppressor that is capable of inducing *p53* expression. Thus, these findings provide additional data on the proapoptotic mechanism of melatonin in cancer cells. Interestingly, after the xenografting of MCF7 cells in rats with an intact circadian rhythm, clock gene circadian expression in MCF7 tumor cells was reestablished [83] (Table 1). Another type of tumor whose risk, progression and aggressiveness are closely related to circadian alterations is prostate cancer. In 2010, Jung-Hynes [84] studied the effect of melatonin (0.1–2 mM) on the expression of different clock genes in prostate cancer cell lines. The study demonstrated that melatonin caused a significant upregulation in *Clock* and *Per2* mRNA and protein levels, as well as a significant downregulation of *Bmal1* mRNA after 24 h of treatment. In addition, melatonin induced a rhythmic oscillation, suggesting that it is able to “resynchronize” the central clock genes in prostate cancer cells [84].

### 5.1. Melatonin and the Ubiquitin–Proteasome System

Several mechanisms have been described to explain the oncostatic effects of melatonin through its effect on clock genes. One of these mechanisms, mentioned above, is the ubiquitin–proteasome system, which enables the degradation of proteins such as BMAL1, REV-ERVα, CRY and PER to be regulated. However, while there is no direct evidence to show that melatonin directly inhibits proteasomes in the SCN, melatonin clearly may interfere with the activity of the ubiquitin–proteasome system in a variety of tissues, including the SCN [88,89]. Vriend and Reiter [76] have suggested that melatonin inhibits the proteasome, which provides selective stability to proteins during periods of elevated melatonin levels. This effect of melatonin on the proteasome results in increased levels of CRY, PER and REV-ERBα, which, in turn, regulate the transcription of *Bmal1*. However, the authors highlight the fact that BMAL1 is closely associated with the inhibition of melatonin in the proteasome, whose expression coincides with the nocturnal increase in this hormone [76]. Moreover, there is even experimental evidence that shows that melatonin inhibits the proteasomes of renal cancer cells (Figure 3) [88].

Currently, it is known that clock genes are closely linked to cell cycle-related genes, whose levels are controlled by the proteasome [90]. Thus, given that melatonin acts as a proteasome inhibitor, its effects on the inhibition of *cyclin D1* and *E*, *CDK2, CDK4* and *c-Myc* expression in brain cancer stem cells [91] or on the reduction of *cyclin D1* and *E* expression in breast cancer cells [92,93] could be closely linked. In addition, in different cancer cells, melatonin and the proteasome inhibitor bortezomib have been shown to have similar proapoptotic effects, indirectly interacting with the levels of p53, Bcl-2, BAX, p21 and NF-kB, through the ubiquitin–proteasome system. Furthermore, other studies have reported that melatonin increases apoptosis due to an increase in the Bcl-2-interacting mediator (BIM) of cell death via the inhibition of proteasome activity [88].

### 5.2. Melatonin and c-Myc

Another important connection between the oncostatic effects of melatonin and circadian machinery is its ability to reduce the levels of c-Myc, which leads to antitumor and antiproliferative effects [91,94]. c-Myc, which is involved in oncogenesis processes, is overexpressed in many cancers. This is explained by the abnormal c-Myc levels, which enable cells with damaged DNA to pass through cell cycle checkpoints, thus contributing to the genesis of many cancers in humans [27,95]. In recent years, this oncogene has been found to control and, in turn, to be controlled by the circadian machinery [95]. Although the circadian connection is currently not entirely clear, MYC is now known to heterodimerize with its partner MAX in order to bind to E-box sequences (CACGTG) at cognate promoters for the purposes of gene regulation, such as the CLOCK-BMAL1 heterodimer [96]. MYC has also been shown to be regulated by CRY and PER proteins [41,97]. Furthermore, downregulation of BMAL1 is associated with high MYC expression levels in a variety of tumors, a phenomenon associated with poor clinical outcomes [98]. Given the multi-faceted involvement, such as increased cell cycle proliferation and increased glycolysis, of *c-Myc* in oncogenesis, as well as the oncostatic effects of melatonin by reducing *c-Myc* expression, we suggest that there is a clear link between this hormone and *c-Myc*, which needs to be studied in more depth (Figure 3).

### 5.3. Melatonin and Sirt1

Melatonin also modulates clock genes through sirtuin 1 (SIRT1). SIRTs are a type III histone/protein deacetylase whose activity depends on oscillating levels of nicotinamide adenine dinucleotide (NAD+). In turn, the biosynthesis or salvaging of NAD+ depends on the enzyme nicotinamide phosphoribosyl transferase (NAMPT), whose expression is characterized by a circadian rhythm [99,100]. Already in 2009, Jung-Hynes and Ahmad [101] proposed that the inhibition of SIRT1 could be the key to the regulation of the circadian machinery. This regulation occurs by controlling the acetylation levels of both BMAL1 and the circadian repressor PER2, thus boosting PER2 degradation and circadian cycle rhythmicity (Figure 3). SIRT1, which is associated with CLOCK, is recruited to the CLOCK:BMAL1 chromatin complex at circadian promoters, thus facilitating the circadian expression of numerous genes, including *nampt*. Therefore, a decrease in NAMPT levels and, chronically, an increase in NAD+ are related to abnormal circadian behavior and metabolism [102,103].

*Sirt1,* which is overexpressed in tumor cells, is correlated with the silencing of tumor suppressor genes and with cancer resistance to chemotherapy. Interestingly, melatonin has a dual effect on SIRT1 expression. In normal cells, melatonin levels increase, leading to a decrease in reactive oxygen species (ROS) production and regulation of cell homeostasis. However, in tumor cells, melatonin downregulates SIRT1, which results in proapoptotic and prooxidant activity [7]. Specifically, in human osteosarcoma, melatonin has been shown to inhibit SIRT1, resulting in increased prooxidant and antitumor activity [104]. The upregulation of SIRT1 by SRT1720 (a known SIRT1 activator) attenuates melatonin’s antioxidant and antitumor activity, indicating that its induction of ROS production in tumor cells is activated by SIRT1 [105]. In addition, in MCF7, Proietti et al. [105] have described how the inhibition of SIRT1 levels by melatonin produces the downregulation of the MDM2 pathway, a ubiquitin protein ligase, which enhances p53 acetylation and, therefore, p53 activity. Although the mechanisms by which melatonin affects SIRT1 are still not fully understood, in 2009, Hill et al. [85] were the first to suggest that melatonin, through the MT1 receptor, blocks the transcription of RORα, thus blocking the activation of *Bmal1* expression and, therefore, of *Sirt1* [85].

### 5.4. Melatonin and AKT

The Akt signaling pathway is involved in the inhibition of cell apoptosis and in stimulating cell proliferation following the activation of Akt/PKB, a serine/threonine kinase. In xenografts of breast cancer, the tumor growth rate and elevated levels of AKT and 3-phosphoinositide-dependent kinase-1 (PDK-1), an AKT stimulator, correlated with night-time light exposure and reduced levels of melatonin [106]. In addition, numerous studies have reported the oncostatic effects of exogenous melatonin due to AKT downregulation [107,108,109]. There is consistent evidence that shows that melatonin may affect the phosphorylation of AKT, PI3K and GSK3β due to the regulation of *Bmal1*. In 2013, Jung, C. H. et al. [110] published a study that showed that the activation of the PI3K-Akt-MMP-2 pathway depends on BMAL1. Specifically, they demonstrated how the inhibition of BMAL1 increased the levels of PI3K activity, AKT phosphorylation, MMP-2 protein and, consequently, cell invasion. An interesting connection has thus been established between the inhibitory effect of melatonin on the AKT pathway and its ability to regulate *Bmal1* (Figure 3) [110].

## 6. Conclusions

Cancer is a complex, heterogeneous disease whose origin is closely related to chronodisruption. Clock genes are directly or indirectly involved in tumor development and in fundamental functions. Despite advances in cancer research, further investigation is required to understand its complexity and to find new therapeutic targets, develop new treatments and/or enhance and improve existing therapies. For many years, the scientific community has reported the numerous oncostatic effects of melatonin, which influences many physiological processes, including the “repair” of the circadian clock. This review describes the possible links between melatonin and the circadian clock in cancer. The mechanisms involved include the ubiquitin–proteasome system and its impact on SIRT1 and c-Myc. Therefore, a more exhaustive study of the effects of melatonin on clock genes could be key to understanding all these mechanisms.

## Figures and Tables

**Figure 1 ijms-24-01919-f001:**
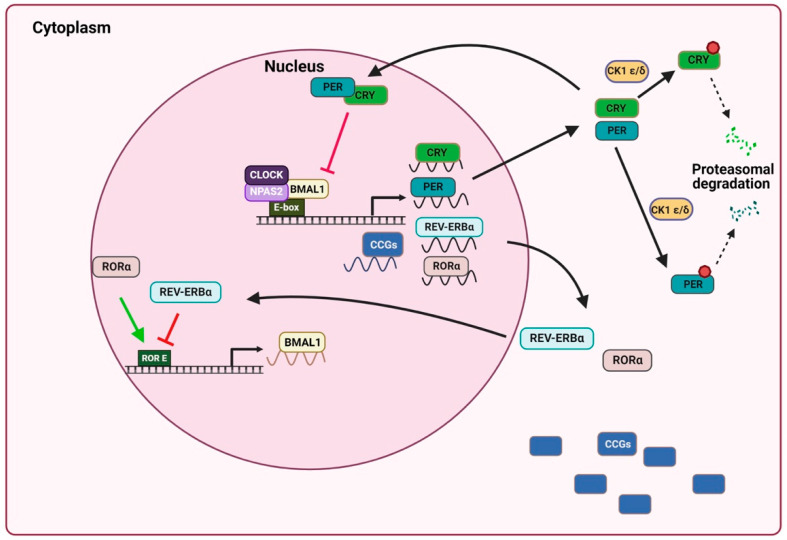
Schematic model of mammalian circadian clock mechanism. The molecular clock is composed of interconnected transcription feedback loops: the transcription factors CLOCK/NPAS2 and BMAL1 produce a heterodimer that binds to the E-box in the promoter and activates the transcription of Per, Cry, Rev-erbα, Rorα and CCGs. CRY and PER dimerize and enter the nucleus, where CLOCK-BMAL1-activated transcription is inhibited, thus generating an oscillatory pattern of gene expression. In the cytoplasm, PER and CRY are phosphorylated by CK1 ε/δ for its degradation. The REV-ERBα receptor inhibits Bmal1 expression, while RORα positively regulates Bmal1 expression. Brain and muscle aryl hydrocarbon receptor nuclear translocator-like protein 1 (Bmal1); cysteine kinase 1 ε/δ (CK1 ε/δ); circadian locomotor output cycles kaput (Clock); cryptochrome (Cry); neuronal PAS domain protein 2 (NPAS2); Period (Per); reverse strand of protein ERB alpha (Rev-erbα); orphan retinoic acid receptor-related alpha (RORα). Image created using BioRender.com (accessed on 22 November 2022).

**Figure 2 ijms-24-01919-f002:**
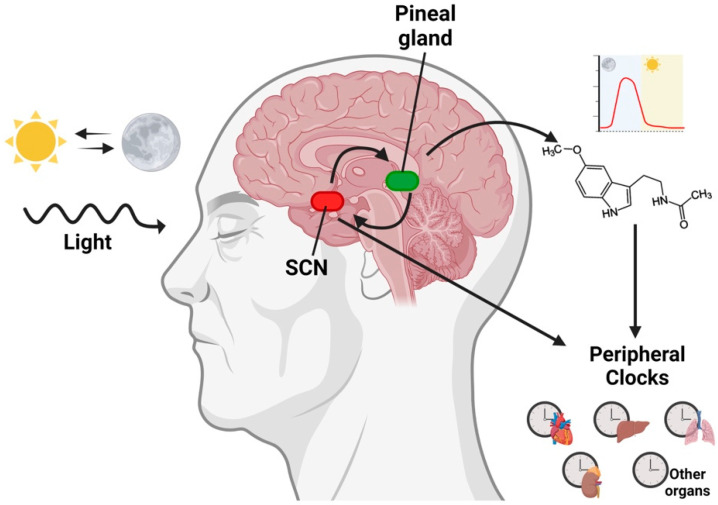
Schematic representation of the role of melatonin in the circadian system. In mammals, circadian rhythms are regulated by circadian clocks. The central clock is located in the SCN of the hypothalamus and controls pineal melatonin secretion in the absence of light, while melatonin also affects the SCN, which regulates chronobiotic activities. In peripheral tissues, clock genes are synchronized by the SCN and are also influenced by melatonin. Suprachiasmatic nucleus (SCN). Image created using BioRender.com (accessed on 22 November 2022).

**Figure 3 ijms-24-01919-f003:**
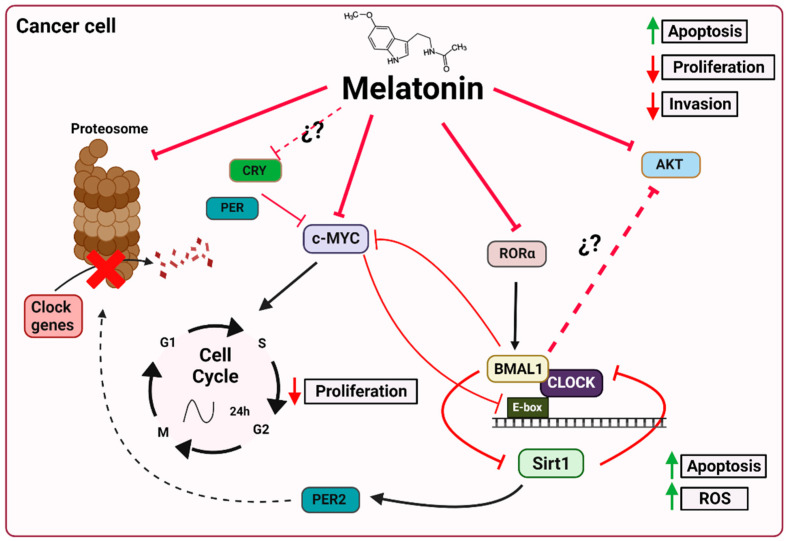
Proposed mechanisms by which melatonin affects the circadian machinery of cancer cells. By inhibiting the proteasome, melatonin can act post-translationally to regulate the circadian clock by stabilizing proteins, depending on the time of day and the levels of this hormone. Melatonin, together with BMAL1, PER and CRY, represses c-Myc, which arrests the cell cycle, thus decreasing cell proliferation. In addition, c-MYC binds to the E-box region and regulates the expression of certain clock genes. Through the MT1 receptor, melatonin blocks the transcription of RORα and also the activation of BMAL1 expression, as well as of Sirt1, with consequent proapoptotic and prooxidant effects. In addition, SIRT1 regulates BMAL1 by controlling acetylation levels and PER2 promoting its degradation. Melatonin inhibits the AKT pathway, which may occur via BMAL1. The inhibition of this pathway causes an increase in apoptosis and a decrease in cell proliferation and invasion. Brain and muscle aryl hydrocarbon receptor nuclear translocator-like protein 1 (Bmal1); circadian locomotor output cycles kaput (Clock); Chryptochrome (Cry); Period (Per); orphan retinoic acid receptor-related alpha (RORα); reactive oxygen species (ROS); sirtuin 1 (Sirt1). Image created using BioRender.com (accessed on 28 December 2022).

**Table 1 ijms-24-01919-t001:** The effect of exogenous melatonin on clock genes in different cancers.

Tumor Type	Study Design	Dose	Results	Author
Breast cancer	In vitro	10 nM	-↓ SIRT1 protein levels	[85]
Prostate cancer	In vitro	100 μM, 1 mM or 2 mM	-↑ *Per2* and *Clock* mRNA and protein levels -↓ *Bmal1* mRNA levels -Rhythmic oscillation of *Per2* and *Dbp*	[84]
Breast cancer	In vitro	1 nM	-↓ *Bmal1* and RORα mRNA levels -↑ *Rev-erbα* and *Per2* mRNA levels at 20 h after treatment	[83]
Hepatocellular carcinoma	In vivo Mice	5, 10 mg/kg	-↓ *Bmal1*, *Clock* and *RORα* mRNA and protein levels -↑ *Per1, Per2, Per3, Cry1, CK1ε* mRNA levels -↓ *Sirt1* mRNA levels -↑ *Rev-Erbα* and *Rev-Erbβ* mRNA and protein levels	[86]
Breast cancer	In vitro	1, 2 mM	Inhibition of LDH-A prevents ↓ Bmal1 under hypoxia conditions	[87]

↓: decrease; ↑: increase.

## Data Availability

Not applicable.

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
