# Peer review of "Role of Melatonin in Cancer: Effect on Clock Genes"

_ijms, 2023, doi:10.3390/ijms24031919_

Round 1

Reviewer 1 Report

First of all, I want to congratulate the authors for this magnificent paper. the review is very complete, dealing with the issues in a clear and orderly manner. There is no doubt about the central role of melatonin in the regulation of circadian rhythms, and its relationship with cancer.

The authors, especially in sections 4 and 5, refer to the role of the MT1 and MT2 melatonin receptors in their physiological effect on clock genes and cancer. Authors should at least introduce a mention of MT3 melatonin binding site, the quinone reductase 2 (QR2), a detoxifying and antioxidant enzyme and its possible implication in the topics discussed. 

Author Response

First of all, I want to congratulate the authors for this magnificent paper. the review is very complete, dealing with the issues in a clear and orderly manner. There is no doubt about the central role of melatonin in the regulation of circadian rhythms, and its relationship with cancer.

The authors, especially in sections 4 and 5, refer to the role of the MT1 and MT2 melatonin receptors in their physiological effect on clock genes and cancer. Authors should at least introduce a mention of MT3 melatonin binding site, the quinone reductase 2 (QR2), a detoxifying and antioxidant enzyme and its possible implication in the topics discussed.

- Corrected.

Page 7: “In addition, melatonin also binds other proteins such as calmodulin, calreticuline and quinone reductase II (previously called MT3), leading to various regulatory effects”

Reviewer 2 Report

Paper by Rodríguez-Santana et al. summarizes nicely up-to-date research on the regulatory system related with the circadian clock and its aberrant functionality resulting in diseases as cancer. In this sense, this review expands the role of the hormone melatonin, as a main regulator of expression of clock genes. The review is really well written and structured, the use of the bibliography is recent with a large number of references, and I think it will be of great interest to scientists studying this field. Therefore, I strongly recommend the acceptance of this manuscript for publication. 

Author Response

Thank you very much for the positive comments.